# Hydrolyzed Rice Formula: An Appropriate Choice for the Treatment of Cow’s Milk Allergy

**DOI:** 10.3390/jcm11164823

**Published:** 2022-08-17

**Authors:** Caterina Anania, Ivana Martinelli, Giulia Brindisi, Daniela De Canditiis, Giovanna De Castro, Anna Maria Zicari, Francesca Olivero

**Affiliations:** 1Department of Mother-Child, Urological Science, Sapienza University of Rome, 00161 Rome, Italy; 2Institute of Applied Calculus-CNR Rome, 00185 Rome, Italy; 3Pediatric Clinic, Department of Pediatrics, Fondazione IRCSS Policlinico San Matteo, University of Pavia, 27100 Pavia, Italy

**Keywords:** cow’s milk allergy, rice, rice hydrolyzed formulas, children

## Abstract

Cow’s milk allergy (CMA) is a common condition in the pediatric population. CMA can induce a diverse range of symptoms of variable intensity. It occurs mainly in the first year of life, and if the child is not breastfed, hypoallergenic formula is the dietary treatment. Extensively hydrolyzed cow’s milk formulas (eHF) with documented hypo-allergenicity can be recommended as the first choice, while amino acid-based formulas (AAF) are recommended for patients with more severe symptoms. Hydrolyzed rice-based formulas (HRFs) are a suitable alternative for infants with CMA that cannot tolerate or do not like eHF and in infants with severe forms of CMA. In the present paper, we reviewed the nutritional composition of HRFs as well as studies regarding their efficacy and tolerance in children, and we provided an updated overview of the recent evidence on the use of HRFs in CMA. The available studies provide evidence that HRFs exhibit excellent efficacy and tolerance and seem to be adequate in providing normal growth in healthy children as well as in children with CMA.

## 1. Introduction

Cow’s milk allergy (CMA) is defined as a reproducible adverse reaction to one or more milk proteins mediated by IgE and/or non-IgE mechanisms [1]. CMA prevalence is reported to be approximately 2% to 3% in the infant population [2,3]. However, this is variable depending on the country and the diagnostic method used. The pan-European birth cohort study using the gold standard diagnostic procedure for food allergies confirmed challenge-proven CMA in 0.54% of children up to two years of age [4]. CMA can induce a diverse range of symptoms of variable intensity [3,5,6]. IgE-mediated reactions, include gastrointestinal, respiratory, and cutaneous symptoms that occur within 1 to 2 h of ingestion [3,7]. Non-IgE-mediated manifestations are less common and mostly involve the gastrointestinal system, including food protein-induced enteropathy (FPE) [8], food protein-induced allergic proctocolitis (FPIAP) [9], and food protein-induced enterocolitis syndrome (FPIES) [10]. CMA occurs mainly in the first year of life, and if the child is not breastfed, hypoallergenic formula is the dietary treatment. Extensively hydrolyzed cow’s milk formulas (eHF) with documented hypo-allergenicity can be recommended as the first choice for the treatment of CMA, especially in infants and young children. Amino acid-based formulas (AAF) can also be recommended, especially for patients with more severe symptoms or in patients not responding to the eHF, and provide effective management for 90% of infants with CMA [3,11]. The efficacy of AA in the CMA treatment has been estimated to be 100% [7]. In some countries, infant formulas based on plant proteins are recommended as a second choice for CMA treatment [5]. Soy- and hydrolyzed rice-based formulas (HRFs) are a suitable alternative for infants with CMA that cannot tolerate or do not like eHF [12]. However, soy formula contains significant amounts of phytoestrogens such as isoflavones, and it is to be avoided in the first six months of life as indicated by the European Society For Pediatric Gastroenterology and Nutrition (ESPGHAN) and European Academy of Allergy and Clinical Immunology (EAACI) [3,11]. Moreover, up to 14% of infants with CMA also react to soy formula [11,12]. HRFs are derived from non-genetically modified rice and do not contain phytoestrogens [13,14]. Several European countries (Italy, Spain, and France) have HRFs available to treat CMA since 2000 [14]. In the present paper, we reviewed the nutritional composition of HRFs as well as studies regarding their efficacy and tolerance in children, and we provided an updated overview of the recent evidence on the use of HRFs in CMA.

## 2. Rice Grain Structure and Nutrient Contents

Rice (Oryza sativa) is a cereal grain belonging to the family of Poaceae (formerly Gramineae or grass) [15]. It represents one of the leading food crops and an important source of protein in the world, and it is cultivated today on every continent. Over 90% of the world’s rice is produced and consumed in the Asia-Pacific region, with China and India alone accounting for more than 50% of global rice production [16]. The mature rice grain is collected in rough rice or paddy rice, in which the caryopsis (or brown rice) is incorporated in a robust siliceous hull (or husk). The caryopsis is the only edible part of rough rice. The hull (or husk) represents 20% of rough rice, whereas the 8–10% is represented by the bran. The bran consisting of the pericarp, seed coat, nucellus, aleurone, pulverized embryo, and some starchy endosperm and hull fragments is removed to obtain milled or white rice composed completely of endosperm [17,18]. The composition and properties of the rice grain are influenced by rice variety, environment, and processing methods [19]. Milled rice is about 78% starch, while protein accounts for 6–7%. Rice bran contains high levels of fat (10–20%) and protein (11–15%), as well as fiber (7–11%). Minerals and vitamins are contained in greater amounts in rice bran than in milled rice [20]. In addition to being considered hypoallergenic, rice protein contained in the caryopsis has a higher biological value and digestibility than that of other grains such as wheat, barley, and corn, but lower than proteins derived from animal sources, legumes, and oil crops [21]. High molecular weight and intermolecular disulfide bonds are responsible for the high water insolubility of rice protein [22,23]. According to the solubility-based classification described by Osborne, rice proteins are classified as follows: albumin (water-soluble), globulin (salt-soluble), glutelin (alkali/acid-soluble), and prolamin (alcohol-soluble) [24]. Although these proteins have a higher quality than other plant sources, such as in the case of soybeans, cereals, or beans, they still have deficiencies in some essential amino acids. Rice lacks three amino acids: lysine, threonine, and tryptophan. In particular, polished rice (endosperm) has a low percentage of lysine and tryptophan, while in brown rice, we have a deficiency of lysine, which makes up only 4 percent of the total weight of the grain. However, the amount of lysine is higher than what can be found in other plant sources, such as in the case of wheat or corn. The biological value of rice protein is higher than that of any other grain: it is 69, compared with 49 in wheat and 44 in corn, for example [14,25]. Rice protein’s water insolubility is the cause of difficult extraction and purification. Currently, alkaline, enzymatic, and physical treatments for the extraction of protein from rice flour have enabled the use of rice proteins as ingredients in nutrition products such as infant formulas. The process of hydrolyzed rice production occurs in two steps: an initial protein extraction step, which is followed by a hydrolysis step. Alkaline extraction facilitates the removal of the endosperm proteins to produce purified starch [22]. Alkaline conditions are particularly effective in extracting proteins from rice flour, where glutenin is the dominant protein fraction [26]. Rice proteins derived by alkaline extraction have higher digestibility and bioavailability compared to those prepared by degradation of starch by α-amylase because of the structural changes caused by alkaline extraction [27]. Enzymatic extraction consists of solubilizing and removing starch by enzymes such as α-amylase, glucoamylase, and pullulanase to obtain isolated proteins. Physical extraction is usually preferred to alkaline and enzymatic methods in food processing because they induce fewer changes [22] as well as generally being more economical and easier to be adapted and utilized in industry. Among the physical treatment, sonication gave the highest protein extraction yield, but this was only 15%. Protein extraction increases when the physical treatment is coupled with enzymatic extraction. Hydrolysis of rice proteins has been shown to yield improvements in terms of solubility, emulsifying, and foaming properties, making the resulting ingredients suitable for a wider range of food applications such as infant formulas [28]. Rice carbohydrates are contained in 34 to 62% in rice bran, 73 to 87% in brown rice, and 77 to 89% in milled rice [29]. Lipids in rice are categorized into starch lipids and no starch lipids; the former are found mainly in the endosperm, while the latter, which represent the most abundant part of the lipids contained in rice, are present in the aleurone, subaleurone, and embryo of brown rice [30]. Bioactive components such as vitamin E are present in rice, and studies have shown that they possess antioxidant, hypocholesterolemic, and antidiabetic activities [31].

## 3. Rice and HRF Allergenicity

Rice allergy is common in Asian regions where this food is commonly used, while the prevalence is much lower in Europe and USA [32,33]. Although rice proteins components are considered generally hypoallergenic, sensitivity to rice proteins is found in 0.7–35% of allergic patients and up to 69% in cereal allergic patients [34,35], but it triggers undesirable reactions in less than 1% of allergic children in Europe [36] and reaches up to 10% in atopic subjects in Japan and India population [32]. IgE and non-IgE mediated symptoms were reported in rice-allergic individuals [37]. Rice allergy is more prevalent in adults than in children. Symptoms frequently associated with rice allergy are atopic dermatitis, eczema, and asthma [37] in sensitized individuals in communities where rice is a staple food, e.g., in the Far East, but is not a frequent cause of food allergy in Western individuals, although the allergy is increasing [33]. Anaphylactic reactions were reported in severe cases [37,38]. Rice is the most common solid food that induces FPIES being the most common trigger in Australia [39]. It is also possible that those who are allergic to rice may also react to other grains, such as barley, corn, oats, wheat, etc., because they are members of the same botanical family [40]. The proteins with molecular masses of 14–16, 26, 33, and 56 kDa were demonstrated to be potentially allergenic [41]. Most of the allergic components are albumins with molecular weights between 14 and 16 kDa [42]. Rice also contains a lipid transfer protein that is heat-stable protein and may account for allergy to cooked rice [43]. Rice has an important aeroallergen (Ory s 1) belonging to the Group I grass pollen allergens. “Rice millers’ syndrome” seems to be associated with exposure to rice husk dust and occurs with acute and chronic irritant effects on the eyes, skin, and upper respiratory tract and allergic-like responses such as rhinitis, dyspnea, bronchospasm, and eosinophilia [44]. However, rice is one of the less allergenic staple foods, reacting in <1% of allergic children in Europe [35]. Moreover, it has no lactose and no phytoestrogens. For this reason, it was developed as a non-allergenic product in rice protein hydrolysates [5]. There are no published cases of reported allergy to HRFs [45]. The allergenicity of a rice hydrolysate (Risolac^®^) was studied in an animal model by Piacentini et al. The author, after a period of sensitization induced by administering either formula with cow’s milk or with HRF, intravenously isolated whole proteins (CMP or rice) or ultra-centrifuged formulas (uCMPF and uHRF). Specific IgG against beta globulin, casein, and whole rice protein was measured. In the group fed cow’s milk protein, intravenous administration of casein or ultracentrifuged whole milk caused more significant allergic reactions than reactions that occurred after HRF administration [46].

## 4. HRF and Cow’s Milk Allergy

Treatment of CMA is dietary treatment of avoiding the intake of cow’s milk and its dairy product. As a substitute for cow’s milk, when breast milk is not available, a therapeutic formula should be used, which guidelines define as such when it is tolerated by at least 90% of children with CMA with a confidence level of 95% [47,48]. HRFs are foods for special medical purposes (FSMP) that meet these criteria, as well as eHF, soy formula, and AAF. Currently, although they are mainly marketed in Italy, Spain, and France, HRF is also available in North Africa, the Middle East, and South America, whereas they are not available in many countries in Europe, the USA, Canada, Australia, and New Zealand [45].

### 4.1. HRFs Composition

#### 4.1.1. Proteins

HRFs are obtained through enzymatic hydrolysis. The rice proteins (80% glutelin and 10% globulin) are insoluble in water, and hydrolysis is necessary to improve water solubility and digestibility [45,49]. In HRFs, peptides possess a low molecular weight [45]. Rice has limited amounts of three essential amino acids in comparison with human milk: lysine, 36 mg vs. 67 mg; threonine, 37 mg vs. 44 mg; and tryptophan, 9 mg vs. 17 mg [48]. Supplementation with free L-lysine, L-treonine, and L-tryptophan is necessary because rice lacks these three amino acids and to meet an infant’s amino acid requirements [14,50]. The nutritional value of a protein is influenced not only by its amino acid composition but also by its digestibility coefficient (DC). The DC of rice proteins is lower than that of CMP (93 vs. 100%). Consequently, the protein content of HRFs is slightly higher than the current average protein content of infant formulas (1.4 g/100 mL), follow-on formulas (1.5 g/100 mL), and growing-up formulas (1.7 g/100 mL) [51].

#### 4.1.2. Lipids

The lipid composition, as well as the energy content of HRFs, is identical to that of standard formulas or follow-on formulas [14].

#### 4.1.3. Carbohydrates

HRFs are lactose-free. The carbohydrate fraction is mainly composed of dextrin–maltose, corn starch, and different kind of syrups [45].

## 5. Tolerance and Efficacy of HRF in CMA

The tolerance and efficacy of HRF in IgE and non-IgE-mediated CMA were evaluated in several studies (Table 1).

In 2003, the study conducted by Fiocchi A. et al. evaluated the tolerance to HRF in 18 children (mean age 5 years) affected by cow’s milk and soy allergy. IgE determination was positive in 7/18 sera for rice; however, a double-blind, placebo-controlled challenge (DBPCFC) with HRF in all the tested patients resulted in being negative. Therefore, they demonstrated that HRF might be used as a protein source for children with multiple food-induced reactions [32]. Subsequently, in 2006, Fiocchi et al. performed a prospective clinical assessment to evaluate the tolerance to an HRF in children allergic to cow’s milk. In their multicenter study, the authors evaluated one hundred children with a diagnosis of IgE-mediated CMA. Sensitization to rice and HRF was investigated, and a DBPCFC was carried out with increasing doses of HRF. Although patients’ sera often contained specific IgE against rice proteins, all DBPCFC with HRF were negative. The authors concluded that HRF is a possible alternative for children with multiple allergies and for those with CMA [52]. In 2010, Reche et al. published a prospective open, randomized clinical study to evaluate the clinical tolerance of a new HRF compared with that of an eHF in the feeding of infants with CMA. Ninety-two infants diagnosed with IgE-mediated CMA were randomized to receive eHF or HRF. The study showed great tolerance to the HRF in infants with moderate to severe symptoms of IgE-mediated CMA, with more than 90% of children developing clinical tolerance. The authors, in accordance with current guidelines, concluded that HRF could provide an adequate and safe alternative to eHF for infants affected by IgE-mediated CMA [53]. Good tolerance in IgE and non-IgE mediated CMA was also demonstrated by Vandenplas et al. in 2014. They conducted a prospective trial for 6 months in 40 infants with IgE mediated and non-IgE mediated CMA, confirmed by oral milk challenge, to study the tolerance and efficacy of HRF. All parameters composing the symptoms-based score (SBS) decreased significantly after 1 month of HRF treatment and remained so after 3 and 6 months [54]. In the same year, Vandenplas et al. published a prospective trial to evaluate the clinical tolerance of a new HRF in thirty-nine infants diagnosed with CMA. Children who entered the study were followed through a SBS. The SBS was significantly lower after 1 month of eHRF feeding than during the challenge (*p* < 0.0001). There was an improvement in symptoms such as hard or watery stools, crying, and regurgitation. After only 1 month of observation, the authors demonstrated clinical tolerance of the HRF (more than 90% of children, 95% CI), with an improvement in the SBS. In addition, in the same children, growth was monitored and evaluated as a z score according to the World Health Organization (WHO) Growth Standards, and a normal weight and length evolution was observed [55].

## 6. Growth Assessment

Some studies have evaluated growth while feeding HRF (Table 2).

Assessment of growth for HRF has occurred in both healthy and children with CMA. The health effects of HRF were investigated in healthy and CMA children. Lasekan et al. and Girardet et al. studied nutritional efficacy in two different studies in which they demonstrated that healthy infants who were fed with HRF from birth until complementary feeding showed appropriate growth, demonstrating the normal nutrition efficiency of these formulas. Lasekan et al. conducted a randomized, blinded, 16-week parallel feeding trial of 65 healthy infants fed either an experimental partially hydrolyzed rice protein-based infant formula fortified with lysine and threonine (HRF, *n* = 32) or a standard intact cow’s milk protein-based formula (CMF, *n* = 33) as a control. They found that weight, length, and head circumference were not different between the two formula groups. Furthermore, all plasma biochemistries for both groups were within the normal reference range. Healthy infants fed an experimental HRF had normal growth, tolerance, and plasma biochemistry comparable to those of infants fed a standard intact milk protein-based formula, despite some differences in amino acid profiles [56]. Girardet et al. conducted an open multicenter prospective study demonstrating that 78 full-term infants fed HRF from the first month of life until 4–6 months of age showed normal growth over the first few months of life, comparable to the WHO standards and a good acceptance of HRF formulas [57]. Five studies evaluated the growth of infants with CMA and fed HRF from the 1st month of life to 2 years of age. In 2003, D’Auria et al. assessed whether HRF allows normal growth and adequate metabolic balance in infants with CMA. All the 16 enrolled infants were randomly assigned to receive HRF (*n* = 8) or a soy formula (control group, *n* = 8). Standardized growth indices (Z scores) and biochemical parameters were evaluated during a 6-month treatment period. Infants in both groups showed normal growth patterns and biochemical parameters without any adverse reactions. The authors concluded that HRF might be a nutritionally suitable alternative for infants with CMA [58]. A clinical trial conducted in 2005 by Savino et al. studied the growth of infants fed with an HRF to assess the nutritional adequacy of this formula. Infants with atopic dermatitis and CMA were enrolled and observed for two years. A total of 88 infants (58 with atopic dermatitis and CMA and 30 with atopic dermatitis without CMA) were studied. The population was divided into three groups: fed with HRF, soy-based formula, or eHF, and a control group of healthy infants fed with a free diet was considered. The authors did not observe any statistically significant differences in weight z-scores among infants fed with HRF, soy-based formula, and eHF during the first 2 years of life, but a significantly lower difference was reported in the HRF group compared to the control group between 9 to 12 months (*p* = 0.025) and between 1 and 1.5 years (*p* = 0.020) of age [59]. In 2007, Agostoni et al. conducted a study investigating differences in growth indices, such as weight-for-age (WA), length-for-age (LA), and weight-for-length (WL), in infants with CMA fed with different types of formula in the complementary feeding period (6–12 months of age). One-hundred and sixty infants (median age 5.3 months) were randomly assigned to soy formula, eHF, and HRF; the control group was made up of allergic infants who were exclusively breastfed. The results demonstrated that all the groups of infants affected by CMA showed positive values of LA z-score gain during the complementary feeding period. However, only infants fed with Hydrolyzed formulas (rice or casein) showed a trend of positive WA z-score gain [60]. In the previously mentioned study conducted in 2010 by Reche et al., growth parameters on ninety-two infants diagnosed with IgE-mediated CMA and randomized to receive eHF or AAF were evaluated. This study, in addition to proving a great tolerance to the HRF by infants with moderate to severe symptoms of IgE-mediated CMA, showed that all children receiving the HRF had similar growth to those receiving eHF. The authors concluded that, in accordance with current guidelines, this HRF could provide an adequate and safe alternative to eHF for infants affected by CMA [53]. In 2014 Vandenplas et al. performed a prospective trial to evaluate the hypo-allergenicity and safety of a new HRF, to assess its validity as an alternative formula in CMA. Thirty-seven infants with CMA confirmed by a food challenge were fed the study formula for 6 months. All infants tolerated the HRF, and the SBS significantly decreased after one month of intervention. At 6 months, all infants showed a normal weight gain as of the first month as well as a normalization of the WA, WL, and BMI z-scores. The authors concluded that HRF is an adequate and safe alternative to eHF [54].

**Table 2 jcm-11-04823-t002:** Main studies evaluating growth while feeding HRF.

Authors, Year, Reference	Type of Study	Number of Subjects	Aim of the Study	Duration of Study	Number of Infants Fed HRF	Number of Infants Fed Another Formula	Outcome
D’Auria et al., 2003, [58]	RCT	16	growth and metabolic balance in infants with CMA-fed HRF	6 months	8	8	normal growth patterns and plasma biochemical parameters in infants fed HRF
Savino et al., 2005, [59]	RCT	88	growth of infants with AD and CMA fed HRF and other formulas	24 months	15	17 SF, 26 eHCF, 30 CG	no difference between HRF and CG, but low weight in HRF raises doubts about the nutritional adequacy
Lasekan et al., 2006, [56]	RDBT	65	Growth, tolerance, plasma biochemistries of infants fed HRF	4 months	32	33	Healthy infants fed HRF showed normal growth, tolerance and plasma biochemistry No difference between the 2 formula groups
Agostoni et al., 2007, [60]	RCT	160	Growth of infants with CMA fed different formulas	6 months	30	32 SF, 31 HCF, 32 CG	HCF and HRF resulted in greater changes in weight for age compared with SF
Reche et al., 2010, [53]	RCT	92	growth of infants with CMA-fed HRF compared with an eHF	24 months	46	46 eHF	children receiving HRF showed similar growth to those receiving an eHF
Vandenplas et al., 2014, [54]	CT	36	growth in infants with a confirmed CMA fed a new eHRF	6 months	36	-	eHRF allowed a catch-up to normal weight gain, a normalization of the weight-for-age, weight-for-length, and BMI z-score

RCT: randomized controlled trial; HRF: Hydrolyzed Rice Formula; AD: atopic dermatitis; SF: soy formula; eHCF: extensively hydrolyzed casein formula; eHF: extensively hydrolyzed formula; HCF: Hydrolyzed Casein Formula; CG: control group; CMA: cow’s milk allergy.

## 7. Taste Acceptability

The palatability of HRF is an important aspect to consider. Given the difficulty of making this assessment in children, many studies were performed on adults. The taste acceptability of 12 different formulas was evaluated in a double-blind study by Pedrosa in 50 randomized adults. The authors demonstrated the superior acceptability of HRF as well as soy formulas over different eHF [61]. Lombardo et al. found that in children, HRF palatability was superior to that eHF [62]. The studies from Fiocchi et al., Lasekan et al., Girardet et al., and D’Auria et al. reported a good acceptance of the HRF, whereas, in the Reche et al. study, two infants out of 46 refused to take HRF [53,56,57,58]. In the study of Vandenplas, 3 of 40 infants enrolled drop-outs due to the bitter taste of the formulas [54].

## 8. Arsenic in HRFs

Current knowledge indicates that organic forms of arsenic (As) have relatively low toxicity, but inorganic As (iAs) is a non-threshold human carcinogen [63]. Rice (Oryza sativa) plants accumulate As more than similar cereal crops. It was suggested that the higher As absorption in rice is due to a high-affinity phosphate/arsenate uptake system [64]. Additionally, the anaerobic paddy soil culture of rice plants also contributes to high As accumulation in rice because phosphate uptake in saturated soil environments does not have the diffusion limitations observed in drier soil [65,66]. Studies of As concentrations in infant rice-based products reported elevated As exposure to infants and young children in many countries [67,68,69,70]. The Food and Drug Administration (FDA) and the ESPGHAN provided recommendations regarding the need to clarify the arsenic content in HRFs [71,72]. Since 2016 The European Union set the maximum rice inorganic arsenic content to 0.10 mg/kg for rice used in the production of food for infants and young children [73]. Meyer et al. studied HRFs in Europe for content on total/iAS and reported that any of the HRF consumed at normal volume (600 mL) intake would equate to an exposure of 0.16–0.23 µg/kg body weight. This is below the average exposure generated from data produced by European Food Safety Authority (EFSA) for both infants (0.24–0.43 µg/kg body weight) and toddler (0.32–0.45 µg/kg body weight µg/kg body weight) and also >10–fold less than WHO guidelines [74]. The authors concluded that the arsenic content is very low in HRF and not different from the arsenic content of cow’s milk-based infant formula [75].

## 9. Conclusions

Available studies provide evidence that HRFs exhibit excellent efficacy and tolerance. HRFs seem to be adequate in providing normal growth in healthy children as well as in children with CMA. Furthermore, more data were published on its safety. In fact, regarding the question concerning the arsenic content in HRFs, it now seems clear that this is very low and that there is no significant difference from the arsenic in cow’s milk formulas. Relative low cost and good palatability are two other aspects that make HRFs useful in the treatment of CMA, especially in the first year of life. Few studies evaluated HRFs role in cases of allergy to hydrolysates and multiple food allergies. In conclusion, HRFs, where available, represent a suitable first-line alternative for the management of CMA.

## Figures and Tables

**Table 1 jcm-11-04823-t001:** Main studies evaluating efficacy and safety of HRF.

Authors, Year, Reference	Type of Study	Number of Subjects	Aim of the Study	Duration of Study	Number of Infants Fed HRF	Number of Infants Fed Another Formula	Outcome
Fiocchi et al., 2003, [32]	CT	18	clinical tolerance to HRF in children with CMA and soy allergy	1 test	18	-	Children with CMA and soy allergy tolerate an HRF clinically
Fiocchi et al., 2006, [52]	PS	100	tolerance to HRF in children with CMA	1 test	100	-	HRF is a possible alternative for children with multiple allergies and CMA
Reche et al., 2010, [53]	RCT	92	clinical tolerance of HRF compared with an eHF in infants with CMA	24 months	41	40 eHF	Children receiving HRF showed similar development of clinical tolerance to those receiving an eHF
Vandenplas et al., 2014, [54]	CT	40	hypo-allergenicity and safety of a new eHRF in infants with a confirmed CMA	6 months	40	-	All infants tolerated the eHRF
Vandenplas et al., 2014, [55]	CT	39	clinical tolerance of a new eHRF in infants with a confirmed CMA	1 month	39	-	eHRF is tolerated by more than 90% of children with proven CMA [95% CI]

CT: clinical trial; PS: prospective study; HRF: hydrolyzed rice formula; CMA: cow’s milk allergy; RCT: randomized controlled trial, eHF: extensively hydrolyzed formula; eHRF: extensively rice hydrolyzed formula.

## Data Availability

Not applicable.

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
