# Peer review of "Hydrolyzed Rice Formula: An Appropriate Choice for the Treatment of Cow’s Milk Allergy"

_jcm, 2022, doi:10.3390/jcm11164823_

Round 1
Reviewer 1 Report
The peer-reviewed paper is the result of a systematic review of papers published in the years 2003-2014 that contain the results of scientific research on the use of Hydrolyzed Rise Formula (HRT) in the dietary treatment of infants with CMA that cannot tolerate or do not like extensively hydrolysed cow's milk protein formulas (eHFs) or amino acid formulas (AAFs) in infants with severe forms of CMA.
The results of Fiochchi’s, Reche’s, Vandenplas' research analyzed by the authors confirm the good tolerance and efficacy of HRF in children with CMA. Based on the results of these studies, the authors conclude that:
· - HRF may be a safe alternative to eHF for children with multiplie food allergies and for a subgroup of children with CMA who do not tolerate eHFs.
· - HRF is very well tolerate by infants with moderate and severe symtoms, both in IgE-mediated and non-IgE-mediated CMA.
The results of D'Auria’s, Savino’s, Agostini’s, Reche’s, Vandenplas’ reasearch assessing growth and metabolic balance in infants with CMA fed HRF, compared with eHF and other milk-free formulas, showed that:
· - all children receiving the HRF had similar growth to those receiving an eHF.
The comparative double-blind study of Petrosa showed:
· - superior acceptability of HRF over different eHFs and other 12 formulas used to treat cow's milk allergic children (evaluation of formulas’ taste, smell and texture).
The study of Meyer confirmed the safety of using HRFs showing:
· - a low arsenic content in this formula, not different from its content in cow's milk based infant formulas.
Based on the analyzed studies, the authors concluded that HRF is characterized by other aspects: relative low cost and good palatability, that make HRFs useful in the treatment of CMA, especially in the first year of life.
The authors also included in this paper very valuable information on the structure of rice grain, biological properties of its components (nutritional, potentially allergenic) and data on allergy to rice proteins (frequency, pathogenetic mechanisms, clinical symptoms) - mainly in adults, in countries where rice is the staple nutrient of the population.
The reviewer shares the authors' opinions that HRFs represents a suitable first-line alternative for the management of CMA. This formula should be available to all children with CMA in various countries of the world, including Poland.
Author Response
RESPONSES TO REVIEWER 1 COMMENTS
(REVIEWER 1)
Thank you for your time and for the positive evaluation of our work.
Best Regards

Reviewer 2 Report
In this review, the authors documented the use of HRFs in CMA, including the nutritional composition of HRFs as well as studies regarding its efficacy and tolerance in children in details. They suggested HRFs exhibit excellent efficacy and tolerance and seem to be adequate in providing normal growth in healthy children as well as in children with CMA. In general, this paper is interesting to readers and well-written. I have no much comments.
Minor comments
1. Table 2. Add Lasekan et al.,2006. (Growth, tolerance and biochemical measures in healthy infants fed a partially hydrolyzed rice protein-based formula: A randomized, blinded, prospective trial. J. Am. Coll. Nutr. 2006, 25, 12–19. )
2. P8,Line 3-4. The sentence in the reference is “a possible explanation for this could be the high level of rice protein hydrolysis, increasing the taste of bitterness.” (Medjad, G.N.; Henocq, A.; Arnaud, B.F. Does the hydrolysis of proteins change the acceptability and the digestive tolerance of milk for infants? The results of a comparative and randomized prospective study. Ann. Pediatr. 1992, 39, 202–206.)
Author Response
RESPONSES TO REVIEWER 2 COMMENTS
(REVIEWER 2)
Point 1. Table 2. Add Lasekan et al.,2006. (Growth, tolerance and biochemical measures in healthy infants fed a partially hydrolyzed rice protein-based formula: A randomized, blinded, prospective trial. J. Am. Coll. Nutr. 2006, 25, 12–19. )
Response 1: we added Lasekan et al., 2006 in the table. However, this was not present in the table because the comparison in this paper is among healthy children, not affected by CMPA.
Point 2. P8, Line 3-4. The sentence in the reference is “a possible explanation for this could be the high level of rice protein hydrolysis, increasing the taste of bitterness.” (Medjad, G.N.; Henocq, A.; Arnaud, B.F. Does the hydrolysis of proteins change the acceptability and the digestive tolerance of milk for infants? The results of a comparative and randomized prospective study. Ann. Pediatr. 1992, 39, 202–206.)
Response 2: we are sorry, but we did not understand if you would like to add that sentence in the text at page 8, line 3-4 (why precisely at that point?), with the reference you suggested. Maybe we did not interpret the suggestion in the right way.
Thank you for your suggestion and comments.

Reviewer 3 Report
It's a very good review that addresses concerns about the use of Hydrolyzed Rice Formula in children with cow's milk allergy. I think it will make a good contribution to this field.
Cow's milk allergy is common in infancy, and advanced hydrolyzed formula or amino acid-based formulas are used in the management of children with cow's milk allergy. however, food rejection is common because these foods do not taste good. Although rice-based formulas taste good, there are concerns that these formulas have equivalent ingredients to milk-based formulas and provide similar growth and development to children fed milk-based formulas. This problem has been addressed in this study. It was a good study on this subject that answered the questions that came to the minds of many physicians. In the review, the problem was handled well, the problem was handled in a clear and understandable way, and it was written very well. There has been a good study on this subject that answers the questions that come to the minds of many physicians, but similar reviews have been reported in the literature recently and do not add anything new to the field on this subject. For example, this creview does not add anything different to the literature than the 14. referrance.
Author Response
RESPONSES TO REVIEWER 3 COMMENTS
(REVIEWER 3)
Thank you for your time and for the positive evaluation of our work. We appreciate the consideration that our work will make a good contribution to this field, considering it as a “niche” subject. Even though there are other reviews about it, as you stated, our aim was to give a more comprehensive review, putting together all the information related to the subject in a single paper, in order for the reader to have the full picture on this matter.
Thank you for your comments and suggestions.
